# Skin Barrier Function and the Microbiome

**DOI:** 10.3390/ijms232113071

**Published:** 2022-10-28

**Authors:** Hyun-Ji Lee, Miri Kim

**Affiliations:** Department of Dermatology, Yeouido St. Mary’s Hospital, College of Medicine, The Catholic University of Korea, #10, 63-ro, Yeongdeungpo-gu, Seoul 07345, Korea

**Keywords:** skin barrier, microbiome, atopic dermatitis, acne, psoriasis

## Abstract

Human skin is the largest organ and serves as the first line of defense against environmental factors. The human microbiota is defined as the total microbial community that coexists in the human body, while the microbiome refers to the collective genome of these microorganisms. Skin microbes do not simply reside on the skin but interact with the skin in a variety of ways, significantly affecting the skin barrier function. Here, we discuss recent insights into the symbiotic relationships between the microbiome and the skin barrier in physical, chemical, and innate/adaptive immunological ways. We discuss the gut-skin axis that affects skin barrier function. Finally, we examine the effects of microbiome dysbiosis on skin barrier function and the role of these effects in inflammatory skin diseases, such as acne, atopic dermatitis, and psoriasis. Microbiome cosmetics can help restore skin barrier function and improve these diseases.

## 1. Introduction

The skin is the outermost surface of the body, with a surface area of about 1 × 8 m^2^. The primary function of the skin is to form a physical, chemical, and immunological protective barrier between the body and the external environment. The skin is composed of stratified keratinized epithelium, which undergoes terminal differentiation to acquire a strong structure. The normal skin surface is an acidic, high-salt, dry, and aerobic environment, but the follicle-sebaceous units are relatively anaerobic and much richer in lipids [1]. The body is interconnected with a complex community of microbes including bacteria, fungi, and viruses that inhabit the body’s surfaces. These microorganisms and their surrounding environment are the microbiome. The commensal microbiome is essential for maintaining skin barrier function by participating in essential physiological processes that occur in the skin [1,2]. In particular, the microbiome participates in physical, chemical, microbial and both innate and adaptive immunological ways in performing skin barrier functions. These interrelationships are the product of a well-controlled and delicately balanced microbiome. However, continuous exposure to various extrinsic and intrinsic factors affects this balanced system, which can lead to pathophysiological problems, inducing inflammatory skin conditions such as infections, allergic diseases, and autoimmune disorders.

In this review, we organized recent studies on the skin microbiome and highlight the latest insights into the role of the microbiome in forming and strengthening the barrier function of the skin. We also illuminated the relationships between the intestinal microbiome and skin barriers. Finally, we examined the effects of microbiome dysbiosis on skin barrier function and its role in inflammatory skin diseases.

## 2. Composition of the Skin Microbiome

The skin microbiome is tremendously diverse in organism number and activity. The composition of the microbiome depends on the physiology of the skin, which is related to a microenvironment characterized by moisture, dryness, and sebaceous content. In addition, the skin microbiome undergoes temporal changes with age [3,4,5]. Finally, the microbiome is influenced by environmental factors, such as diet, antibiotic use, and obesity [4,6]. In this section, we discuss the composition of the microbiome of healthy individuals.

### 2.1. Site-Specific Composition of Microbiome

Recent sequencing studies have extensively mapped the species inhabiting the skin of various body parts. These studies have shown that the skin of each body part, driven by the hair follicle and gland composition resulting in distinct niches for bacterial growth, exhibits different microbe compositions. Sebaceous sites, such as the glabella and back, are dominated by lipophilic *Cutibacterium* species, which are closely associated with acne vulgaris [4]. Moist sites, such as the bends of the elbows and knees and toe webs, are largely colonized by *Staphylococcus* and *Corynebacterium* species and beta-*Proteobacteria.* Dry sites, such as volar forearm, are mainly colonized by *Cutibacterium acnes* (*C. acnes*) and *Corynebacterium* species [4].

In addition to bacteria, which are the most abundant microbial organisms on the skin, numerous fungal and virus communities inhabit the skin. The species of fungi differ in composition depending on the body part, not the physiological condition of the skin [5,6]. The *Malassezia* species predominates on the core body and arms, whereas the foot is colonized by a much greater fungal diversity, including *Malassezia* spp., *Aspergillus* spp., *Cryptococcus* spp., *Rhodotorula* spp., and *Epicoccum* spp. [5]. In contrast to bacteria and fungi, eukaryotic DNA viral composition, predominantly Polyomaviridae and Papillomaviridae, is specific to the individual rather than physiological state or anatomical site [3].

### 2.2. Temporal Changes of the Composition of the Microbiome

Skin microbe composition also experiences temporal changes throughout the life cycle. Formation of the skin microbiome is presumed to start in the fetus. The presence of DNA in the microbiome, such as that of *Cutibacterium* and *Staphylococcus* species, has been mapped in amniotic fluid [7]. During normal delivery, the neonate’s skin is exposed to the microbes of the birth canal. Next, the neonate’s skin microbiome is influenced by the interaction with the external environment. One study found that a neonate’s skin microbiome is strongly influenced by the mother’s microbiome at birth, and that there are significant differences in both the skin and gut microbiomes between babies born naturally and those born by cesarean section [8,9]. A study by Maria G D. et al. revealed that using multiplexed 16S rRNA gene pyrosequencing, vaginally delivered infants acquired bacterial communities similar to maternal vaginal microbial communities dominated by *Lactobacillus*, *Prevotella*, or *Sneathia* spp., while C-section infants have skin bacterial communities resembling those found on the normal skin surface, dominated by *Staphylococcus*, *Corynebacterium*, and *Cutibacterium* spp. [8]. At puberty, the skin microbiome undergoes a major shift as sex hormones are secreted, driving sebaceous gland maturation and sebum production, transitioning toward a more adult microbiome. Lipophilic *Cutibacterium* species among bacteria and *Malassezia* among fungi predominate, with decreasing diversity during this period. *Staphylococcus* and *Streptococcus* species, which are more prominent in female children, gradually decrease [10].

After puberty, despite continuous exposure to the external environment and individual factors that are imposed on the skin, the microbial composition of the skin remains relatively stable within an individual over time [3]. The stability of these microbiomes is maintained by mutually beneficial symbiotic relationships between microorganisms and between microorganisms and the host. The relationship between *C. acnes* and pilosebaceous units play an important role in this stability. The pore structure of the pilosebaceous unit and its anaerobic environment provide a random single-cell bottleneck for *C. acnes* and enables stable colonization [11].

## 3. Skin Microbiome and Barrier Function

The skin microbiome and the skin barrier relate symbiotically to affect each other in physical, chemical, and immunological ways. The microbiome also interacts directly with pathogenic microorganisms encountered on the skin surface. In this section, we discuss microbial fortification of the skin barrier (Figure 1).

### 3.1. Physical Barrier

The physical barrier of the skin is the first barricade against external invasions. This barrier is formed by multiple fine layers of epidermal keratinocytes and undergoes a tightly controlled terminal differentiation to form the stratum corneum, which is influenced by the microbiome. Epidermal keratinocytes maintain close physical contingence to each other through a tight junction and adherens intersections, forming a nearly impermeable protective layer against pathogenic microorganisms. In addition, tight junction proteins, such as the zona occludens, can contribute to proliferation and differentiation of keratinocytes in the wound healing process, reconstructing the barrier to microorganisms [12]. Recent studies have shown that using germ free mice, the microbiota is essential for the epithelial barrier integrity and function of the skin barrier. These functions are mediated by the aryl hydrocarbon receptor (AHR) of keratinocytes. Mice lacking AHR are more vulnerable to barrier damage and infection. In skin damage, microbes produce metabolites that activate the AHR in keratinocytes, promoting epithelial differentiation and supporting epithelial integrity [13]. Another role of the microbiome is to secrete the components that make up the lipid structure. Using the mouse model, a study reported that *S. epidermidis* secretes a sphingomyelinase that helps the host to acquire essential nutrients for the bacteria and produce ceramide, a key component of the epithelial barrier that prevents skin dehydration and aging. In this study, *S. epidermidis* significantly increases skin ceramide levels and prevents water loss of damaged skin in a manner that is entirely dependent on sphingomyelinase [14].

### 3.2. Chemical Barrier

The chemical barrier of the skin is organized by numerous lipids and acids secreted by the epidermis and the microbiome. The microbial composition is similar in most areas of the skin but differs among sebaceous, moist and dry areas [2]. Both *C. acnes* and *Corynebacterium*, enriched in sebaceous regions, secrete lipases that hydrolyze free fatty acids from triglycerides in sebum [15,16]. Free fatty acids maintain a low pH, inhibiting the growth of pathogenic microorganisms. In addition, free fatty acids not only serve to directly inhibit bacteria, but also enhance skin immunity by stimulating the expression of human β-defensin 2 (hBD-2), one of the most abundant antimicrobial peptides (AMP)s in human skin [17]. The acidity of the stratum corneum is important for both permeability barrier formation and antimicrobial defense. The acidic skin surface creates a chemical environment that is hostile to pathogenic bacterial colonies. In addition, pathogenic microorganisms are directly inhibited by some lipids or free fatty acids. For example, sapienic acid from the stratum corneum can effectively inhibit pathogenic *Staphylococcus aureus* (*S. aureus*), but this acid does not have sufficient activity against *Staphylococcus* and *Corynebacterium*, which are major components of the skin microbiome [18]. Overall, the composition and function of the microbiome are complementary to the chemical barriers formed by lipids and fatty acids in the skin.

### 3.3. Innate Immune Barrier

In addition to the aforementioned barriers, the microbiome also stimulates a range of innate immune responses and maintains symbiotic relationships with the skin when the skin barrier is disturbed. For example, *S. epidermidis* modulates the innate immune system by activating γδ T cells and upregulating perforin-2, an AMP with unique properties against intracellular pathogens [19]. In addition, a specific glycan expressed in *S. epidermidis* is required for homeostatic T cell activation of *S. epidermidis* by interaction with C-type lectin in human immune cells [20]. Similarly, *Candida albicans*, which is a dimorphic fungus causing mucocutaneous and systemic infections, can stimulate T helper (Th) 1 or Th17 immune responses in the skin, protecting against cutaneous or systemic infection [21].

The skin microbiome modulates the production of a variety of innate factors, including interleukin 1a (IL-1α) [22], components of complement C5a receptor [23], and AMPs derived from keratinocytes and sebocytes, to enhance the innate immune system through a variety of mechanisms [24,25,26]. Representatively, AMP LL-37 (the active form of which is cleaved from the protein cathelicidin) is increased in response to the activation of Toll-like receptor (TLR)-2 signaling initiated by small molecules produced by *S. epidermidis* [27]. Stimulation from the commensal microbiome also induces members of the β-defensin family with bactericidal action against *Escherichia coli* and *S. aureus* from the skin [28]. Some functional keratinocytes of the skin appendages, such as sweat glands, sebaceous glands, and hair follicles contain various AMPs associated with a specific intrinsic microenvironment. While dermcidin may be a sweat gland-specific AMP, new evidence has suggested that this protein is also produced by the sebaceous glands of humans and mice [29].

The relationship between the host and the microbiome may vary depending on the oxygen content in site-specific skin areas and the resulting distribution of the microbiome. The facultative anaerobic bacterium *C. acnes* undergoes fermentation and generates short-chain fatty acids (SCFAs). *C. acnes*-derived SCFAs inhibit histone deacetylases (HDAC)-8 and -9 and promote the activation of fatty acid receptors, which induce activation of cytokines via the TLR-2 or TLR-3 signaling pathway [30,31]. Additionally, SCFAs can limit biofilm formation by *S. epidermidis* [32]. Hence, microbial metabolic and inflammatory backgrounds can trigger the unique characteristics of immune responses.

In addition, when the sebaceous gland is exposed to the gram-negative bacterial cell wall component lipopolysaccharide, the gland produces bactericidal molecules and small proline-rich proteins (SPRR)1 and SPRR2. Human SPRR has strong bactericidal activity that directly disturbs the pathogen’s negatively charged membrane [33]. Commensal bacteria can also act on lipids secreted from sebaceous glands and hydrolyze these lipids to free fatty acids (FFAs) [34]. FFAs have a unique antibacterial effect against various gram-positive bacteria and can induce sebaceous cells to upregulate the expression of hBD-2 [17]. Among these, sapienic acid has bactericidal activity against methicillin-resistant *S. aureus* (MRSA) [35].

### 3.4. Adaptive Immune Barrier

The relationship between adaptive immunity and microorganisms is especially important in developmental processes. This relationship supports the proposition that much of the complex’s adaptive immune system acts to maintain symbiosis with these microorganisms. Regulatory T cells (T regs) and innate-like/unconventional cells play an important role in this immunity. In a mouse model, exposure to the skin commensal *S. epidermidis* induced the abrupt accumulation of T regs on neonatal skin [36]. In these experiments, disruption of T reg responses to *S. epidermidis* in neonates resulted in barrier disruption, increasing inflammatory response upon secondary exposure to commensal microbes. In addition, the accumulation of T regs arises with coordination between hair follicle morphogenesis, which acts as the first reservoir of microbes, and the pathway of chemokine Ccl20-Ccr6 [37].

Mucosal-associated invariant T (MAIT) cells, along with invariant natural killer T cells and γδ T cells, are a type of evolutionarily archaic unconventional T cell that recognizes conserved antigen sets and molecules restricted by major histocompatibility class Ib (MHC-Ib) [38]. As MHC-Ib molecules can present antigens with specific chemical or amino acid sequence motifs, unconventional T cells under their control may be the most effective cells for detection and recognition of antigens and metabolites derived from the microbiome. MAIT cells are activated by MHC class I-like molecules, present in numerous commensal bacteria and yeasts, but not in mammalian cells. The MHC class I-like molecules include a folic acid (vitamin B9) metabolite and riboflavin (vitamin B2) [39]. Within the human skin, MAIT cells comprise a significant portion of lymphocytes, accounting for approximately 2% of CD3+ lymphocytes [40].

Skin microbes do not play as important a role as those of the intestine in optimal seeding of dermal T regs [22]. Even so, the continued action of T regs in the skin modulates responsiveness to symbionts after adulthood, as in the intestine. For example, in experiments using mouse models, commensal-specific T cells produce abnormal type 2 cytokines when there is a specific defect in T reg fitness in the skin. This may be the etiology of several inflammatory skin diseases [41]. A recent study reported that exposure to *S. aureus* in neonates increased IL-1β in response to *S. aureus*-associated α-toxin, which actively inhibited the induction of T regs [42]. Collectively, these findings suggest that microbes are critical in the construction and activation of immune cells.

### 3.5. Microbial Barrier

In addition to microbe-host relationships, microbe–microbe relationships act as a barrier against invasion, colonization, and infection by invading, pathogenic, or opportunistic microbes. The interactions maintain survival and sustain microbes’ niche and access to nutrients. While these relationships are not well understood, regulation of the coagulase-negative *Staphylococcus* (CoNS) species for *S. aureus* has been well studied [43]. CoNS species are among the most abundant microbes in the skin microbiome, including *S. epidermidis*, *S. capitis*, *S. caprae*, *S. hominis*, *S. lugdunensis*, and *S. haemolyticus*.

The most representative way in which CoNS inhibits *S. aureus* is through secretion of AMPs. For example, some strains of *S. hominis* secrete antibiotics in patients with atopic dermatitis (AD), inhibiting *S. aureus* colonization [32]. Additionally, *S. lugdunensis* in the nasal cavity produces lugdunin, a potent anti-*S. aureus* AMP. Lugdunin activates keratinocytes to release LL-37 and a chemoattractant, CXCL8/MIP-2, resulting in recruitment of neutrophils [44,45]. Phenol-soluble modulin secreted by various CoNS species directly antagonizes *S. pyogenes*, *S. aureus*, and *C. acnes* and enhances antimicrobial activity in cooperation with the host keratinocyte-derived AMP LL-37 [46,47]. Additionally, in a pilosebaceous unit, *C. acnes* secrete a competitive thiopeptide antibiotic, cutimycin, to maintain its niche against *Staphylococcus* species colonization [48].

In addition to AMPs, all staphylococci have an inter-microbial communication system called quorum sensing that operates through an auxiliary gene regulator (agr) system [49]. This system is required for *S. aureus* skin infection, colonization, damage, and inflammation, providing a target for commensal microbes to inhibit the growth and toxicity of pathogenic microbes. *S. caprae* showed improvement of infection by inhibiting methicillin-resistant *S. aureus* agr activity [50]. *S. hominis* and other CoNS interfered with the quorum sensing system of *S. aureus*, reducing toxin production and tissue damage and inflammation in an AD model [49].

Production of proteases is another effective mechanism by which CoNS species interfere with *S. aureus* toxicity. A subset of *S. epidermidis* can produce a serine protease that inhibits biofilm formation and colonization of *S. aureus* [51]. *Corynebacterium accolens* release corynebacterial lipase, which restricts the growth of *S. pneumoniae* [15]. Collectively, the competitive relationship between these microbes plays an important role in maintaining the balance of the skin microbiome.

## 4. Gut-Skin Axis and Skin Barrier Function

Cumulative evidence has demonstrated that skin and other barrier sites, like intestine, lung, and brain, have a bidirectional connection. Especially, the intestine and skin are both highly innervated and vascularized and interface with the external environment. The immune systems of both organs are consistently activated for homeostatic conditions, called the gut-skin axis [52]. In this regard, skin conditions can affect the intestinal microbiome. For example, exposure to narrow band ultraviolet (NB-UVB) light can inflect the intestinal microbiome. A study reported that after exposing the skin to NB-UVB, fecal microbiota composition analysis using 16S rRNA sequencing significantly increased alpha and beta diversity [53]. In this study, they also revealed that the serum 25(OH)D concentrations, which is associated with exposure to sunlight/UVB light, correlated with the relative abundance of the *Lachnospiraceae*, *Lachnopsira* and *Fusicatenibacter* genera [53]. Also, food allergies can be caused by exposure to epidermal protein in household dust, ultimately leading to immunoglobulin E-mediated mast cell expansion in the intestine [54]. Recent studies reported that scratching in a mouse model caused keratinocytes to release IL-33. This led to type 2 innate lymphoid cell secretion of IL-4 to activate mast cells of the small intestine that increases intestinal permeability and food anaphylaxis [55]. Another study suggested that the skin–gut correlation depends on the intestinal stromal fibroblast. In a mouse model of inflammatory bowel disease, wounding of the skin activated hyaluronan catabolism, resulting in alteration of colon fibroblast function, and subsequently increased inflammation in the colon by the production of AMPs and alteration of the fecal microbiome [56].

The gut microbiome can also have a significant impact on the overall homeostasis of the skin. The integrity of the intestinal barrier, along with the action of intestinal mucus, immune cells, immunoglobulin A, and AMPs, prevents the entrance of bacteria into the bloodstream, ultimately maintaining skin homeostasis. Through its influence on the signaling pathways that coordinate this process that is essential to skin homeostasis, the gut microbiome impacts integumentary health [57]. According to recent studies, the gut microbiome appears to be involved in skin barrier function by controlling the modulatory effect on systemic immunity. Retinoid acid include: Polysaccharide A, a metabolite of the gut microbe *Bacteroides fragilis*; *Faecalibacterium prausnitzii*; and *Clostridium* species that activate the accumulation of T regs [58]. Another important metabolite, SCFA, is also produced by gut microbes; this metabolite executes anti-inflammatory activities in the skin. Segmented filamentous bacteria stimulate pro-inflammatory Th17 and Th1 cells [59,60]. Additionally, a study with a psoriasis mouse model found that enhanced Th17 inflammation is promoted by the intestinal microbiome [61]. Furthermore, the gut microbiome and its metabolites may directly metastasize and affect the skin barrier directly. O’Neill et al. reported that gut microbes and their metabolites can be absorbed systemically when the intestinal barrier is disrupted in psoriatic patients and reach the skin and affect skin barrier function [57].

The constructive effects of intestinal bacteria supplementation on skin barrier function have been suggested in previous studies. In a study by Levkovich et al., mice receiving *Lactobacillus reuteri* supplementation showed increased dermal thickness, increased hair follicle formation, and increased sebocyte production compared to the control group [62]. In human clinical studies conducted by Ogawa et al. and Guéniche et al., oral supplementation with *Lactobacillus* induced a significant reduction in transepidermal water loss (TEWL), an indicator of skin barrier function. The reduction of TEWL has been shown to be due to an increase in circulating transforming growth factor beta (TGF-β), a cytokine important for maintaining skin barrier function [63,64].

The intestinal microbiome also contributes to repair damaged skin barrier function through both innate and adaptive immunity processes [65,66]. For example, several studies using an atopic dermatitis mouse model showed that oral supplementation with *Lactobacillus* increased general skin barrier function, including decrease in severity of disease, subsequent TEWL, and tumor necrosis factor alpha (TNF-α) release [67,68]. The gut microbiome may also have an effect on wound healing. Supplementation with lactobacillus decreased the time required for wound healing by triggering oxytocin to activate host T regs and rapidly remove neutrophils in wound sites [69].

## 5. Inflammatory Skin Disorders and Microbiome

### 5.1. Atopic Dermatitis (AD)

The microbial composition in skin with AD lesions is characterized by an increased abundance of *S. aureus* and reduced diversity compared with healthy skin and skin without lesions. Patients with more severe AD have decreased alpha- and beta-diversity on the skin. These decreases in diversity can also be seen with acute exacerbations of AD, with reductions in the genera *Streptococcus*, *Corynebacterium*, and *Cutibacterium* and the phylum *Proteobacteria* toward the genus *Staphylococcus* in general and *S. aureus* in particular [70]. *S. aureus* can overwhelm the symbiotic microbiome and induce exacerbation of AD [71]. *S. aureus* interferes with the host immune response, directly damage the skin barrier, and impair adaptive immunity. This is the result of alpha toxin production that causes IL-1R-mediated inflammation and limits the accumulation of *S. aureus*-specific T regs [42]. Therefore, changed microbiome metabolites in atopic skin can cause inflammation and maintain and worsen barrier dysfunction. In contrast, CoNS is usually dominant in the non-lesion skin of AD [72]. Nakatsuji et al. have shown that CoNS strains isolated from the skin of healthy individuals have a better ability to kill *S. aureus* compared to CoNS strains isolated from AD skin [73]. At least 10 proteases from *S. aureus* in AD lesional skin can facilitate dissolution and penetration through the stratum corneum. Additionally, *S. aureus* strains isolated from AD patients possess different surface proteins compared to common strains. For instance, Proinflammatory staphylococcal lipoproteins induce thymic stromal lymphopoietin expression in a TLR2/TLR6-dependent manner. These proteins affect skin adhesion and induce imbalanced Th1/Th2 adaptive immune responses via Langerhans cells [74].

Emerging studies have demonstrated that the gut–skin axis can be associated with allergic diseases including AD, supporting the “microbiome hypothesis” [75]. A metagenomic analysis of fecal samples from patients with AD showed a considerable decrease in *Faecalibacterium prausnitzii* species and SCFAs produced by *Faecalibacterium prausnitzii* compared with the controls. Data indicated the presence of a possible positive feedback loop. The altered gut microbiome allows penetration of poorly digested food, microbes, and toxins, and is associated with epithelial barrier dysfunction in AD tis patients. The altered microbiome triggers Th2 inflammation, resulting in further disruption of skin barrier functions [76]. Overgrowth of multiple *Malassezia* spp. (*Malassezia furfur* and *Malassezia sympodialis*) may cause AD pathogenesis [77]. A high proportion of AD patients present with a positive reaction to Malassezia allergens. There is increased levels of Malassezia-specific IgE in AD patients and a correlation of AD severity [78]. In particular, *M. sympodialis* can release extracellular vesicles, inducing IL-4 and TNF-α. *M. sympodialis* can also induce leukotrienes in IgE-sensitized mast cells and enhance IgE-mediated degranulation of mast cells.

Recently, probiotics and prebiotics have been studied as innovative treatments for skin diseases such as AD [79,80,81]. The results of the studies are still controversial. While some studies suggest improved symptoms, quality of life, and the clinical severity of AD, others have not been able to confirm these findings. The impact of topical treatment on the skin microbiome and consequent influence on AD have been explored in multiple studies. Studies have shown that long-term use of emollients increases the proportion of *S. salivarius* in infants at a high risk of AD [82]. Emollient-mediated microbiome changes can play a role in correcting the imbalance of the skin microbiome associated with AD and its prevention. In addition, several studies have shown that the use of microbiome-supplemented emollients can improve the symptoms and severity of AD compared to normal emollients. The strains tested in these studies are *Vitreoscilla filiformis* [83], *Aquaphilus dolomiae* [84], *Lactobacillus reuteri* [85], and *Roseomonas mucosa* [86]; these species increase the proportion of CoNS in the skin and inhibit colonization by *S. aureus*. The benefits of their use are decreased in pathogenic microbes, reduced production of toxic metabolites, and increased homeostasis of the skin barrier function.

### 5.2. Psoriasis

As mentioned in the previous section, the microbiome plays an important role in the Th17 immune response. The Th17 inflammatory pathway is key in the pathogenesis of psoriasis, suggesting that the microbiome plays an important role in the pathogenesis of psoriasis. However, previous studies on the microbiome in psoriasis have had inconsistent results, and difficulty arises in drawing a conclusion about the relationship between psoriasis and the microbiome [87]. Diverse studies of the microbiome of patients with psoriasis have reported controversial results on trends in *Firmicutes*, *Actinobacteria*, and *Proteobacteria*. These studies did not draw convincing conclusions on the diversity of the microbial community in skin affected by psoriasis lesions compared to that of healthy skin. In psoriatic lesions, abnormal colonization of *S. aureus* and a decrease in *S. epidermidis* abundance were consistently observed, although overt infection is rare [88,89]. Chang et al. reported that mice colonized with *S. aureus* demonstrated a strong Th17 polarization compared to mice colonized with *S. epidermis* [90]. Their results suggest that *S. aureus* can lead to Th17 inflammation observed in psoriasis.

The association between gut dysbiosis and psoriasis is relatively well understood. Host- and microbiome-derived factors lead to gut dysbiosis, resulting in a low-grade chronic inflammatory process [91]. For example, changes in gut microbiome composition can cause changes in levels of SCFAs, produced by Firmicutes and Bacteroidetes. It enhances Th17 inflammation, promoting psoriasis-like skin inflammation. Additionally, these factors cause a decrease in T reg levels in psoriasis patients, resulting in an imbalance between effector T cells and suppressor T cells [92]. Phenol, a metabolite of aromatic amino acids from a disturbed intestinal environment and considered a bioactive toxin, affects abnormal skin keratinocyte differentiation through unknown mechanisms [93]. Further, in a study with a psoriasis mouse model, *C. albicans* exposure enhanced Th 17 inflammation through IL-17 production by αβ T cells sensitized by *Candida* [94].

### 5.3. Acne

While *C. acnes* is a major commensal organism in the normal skin flora, this bacterium contributes to the pathogenesis of acne [95]. Previous metagenomics studies have found similar proliferation levels of *C. acnes* between acne patients and healthy individuals, with a slightly higher level reported in healthy subjects [96,97]. Instead, loss of microbial diversity and of balance between *C. acnes* phylotypes has been reported in acne patients; these may be important etiological factors in the pathogenesis of acne [95]. Moreover, the severity of acne is associated with a loss of diversity of *C. acnes* phylotypes rather than proliferation of *C. acnes* [98]. Loss of diversity is characterized by colonization of certain strains of *C. acnes*, particularly acne-associated phylotype IA_1_, which is enhanced by a hyper-seborrheic environment [99]. Loss of *C. acnes* phylotype diversity can activate the innate immune system and skin inflammation in acne. Indeed, monoculture of phylogenetic IA_1_ has been shown to induce activation of the innate immune system [100]. Additionally, *C. acnes* strains can differentially modulate the CD4+ T-cell responses, activating Th 17 cells and inducing production of interferon gamma [101].

A recent study reported the importance of the relationship of *S. epidermidis* and *C. acnes* in maintaining normal skin barrier function [102,103]. Succinic acid, a metabolite of *S. epidermidis*, inhibits the growth and colonization of *C. acnes* and the inflammatory response induced by *C. acnes* [104]. In addition, *S. epidermidis* can inhibit the *C. acnes*-induced keratinocyte production of IL-6 and TNF-α [103].

In the previous section, we noted that the gut microbiome can affect skin barrier function. This phenomenon is also seen in acne patients. Previous studies showed that the gut microbiome composition was different between moderate to severe acne patients and controls. The patients had less abundant *Actinobacteria* and more abundant *Proteobacteria* [105]. In another study, the ratio of *Bacteroidetes* to *Firmicutes* was higher in acne patients; this was interpreted as an effect of the Western diet [106].

As in AD, research on treatment and management using probiotics and microbiome cosmetics in acne is ongoing. In one study, applying *S. thermophilus* to the skin increased the production of ceramides, a major skin lipid with anti-inflammatory and antibacterial effects against *C. acnes* [107,108]. Another study showed that application of emollients with *E. faecalis* reduced the severity of the disease in patients with mild to moderate acne [109].

The possible mechanisms of the microbiome on each inflammatory skin disease is summarized in Table 1.

## 6. Conclusions

As academic interest in the microbiome increases, research on the effect of the microbiome on skin barrier function is being actively conducted. In this review, we discussed the interaction between the microbiome and skin barrier function. Microbiome dysbiosis can cause damage to the skin barrier, which is associated with inflammatory skin diseases. Recently, studies on the positive effects of oral or cosmetic microbiome supplementation on skin diseases have been conducted. Microbiome supplementation may help restore skin barrier function and improve disease, which may open new directions for the treatment of inflammatory skin diseases.

## Figures and Tables

**Figure 1 ijms-23-13071-f001:**
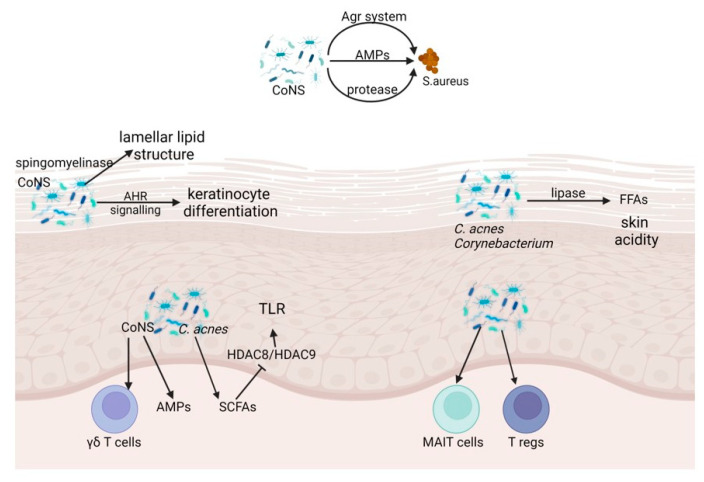
Microbial, physical, chemical, and innate and adaptive immune barrier function of skin microbiome. CoNS, *Coagulase negative Staphylococcus*; Agr, auxiliary gene regulator; AMPs, antimicrobial peptides; AHR, aryl hydrocarbonate receptor; *C. acnes*, *Cutibacterium acnes*; SCFA, short-chain fatty acids; HDAC, histone deacetylases; MAIT, Mucosal-associated invariant T; T regs, regulatory T cells.

**Table 1 ijms-23-13071-t001:** Inflammatory skin disorders and microbiome.

Diseases	Strains	Molecules from Strain	Mechanisms
Atopic dermatitis	*S. aureus*	alpha toxin	IL-1R-mediated inflammation limits the accumulation of *S. aureus*-specific regulatory T cells [42]
		proteases	facilitate dissolution and penetration through the stratum corneum [73]
		surface proteins	induce thymic stromal lymphopoietin expression [74]
	*Malassezia* spp.	extracellular vesicles	induce IL-4 and TNF-α induce leukotrienes [77]
Psoriasis	*S. aureus*	-	strong Th17 polarization [90]
	gut dysbiosis	SCFAs	enhance Th17 inflammation and decrease in regulatory T cell levels [92]
		Phenol	affects abnormal skin keratinocyte differentiation [93]
	*C. albicans*	-	Sensitize αβ T cells and produce IL-17 [94]
Acne	*C. acnes*	Acne-associated phylotype IA_1_	induce activation of the innate immune system [99] modulate the CD4+ T-cell responses, activating Th 17 cells and inducing production of interferon gamma [101]

*S. aureus*, *Staphylococcus aureus*; IL, interleukin; TNF-α, tumor necrosis factor alpha; Th, helper T cell; SCFAs, short-chain fatty acids.

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
