# Peer review of "Skin Barrier Function and the Microbiome"

_ijms, 2022, doi:10.3390/ijms232113071_

Round 1
Reviewer 1 Report
This article is impressive because of creative insights into the symbiotic relationships between the microbiome and skin barrier. This article is well structured article showing the effects of microbiome dysbiosis on skin barrier function and the role of microbiome in inflammatory skin diseases. It would be better if some contents were modified.
1. Figure 1 is a great figure showing the summary of third topic, “skin microbiome and barrier function”. However, since it does not apply to all bacteria, it would be helpful to write down the corresponding bacteria as described in the microbial barrier.
2. This article demonstrates that microbiomes maintain a low pH and stimulate the expression of AMPs by secreting lipases that hydrolyze free fatty acids. It would be better if the description in figure 1 is more detailed in accordance with this content.
3. The 5th topic, “Inflammatory skin disorders and microbiome”, is good content describing the relationship between diseases and the microbiome. However, It would be better if you summarize the contents in a table and describe which species have which mechanisms and how they affect diseases.
4. In the 5th topic, the readability would be better if you add 1 blank line between the sub-paragraphs by diseases.
5. In line 346-356, the article describes the association between psoriasis and microbiome through Th17 immune response. However, unlike other diseases, it seems to be less mention of specific species. It would be better to mention the species if specific species are involved in Th17 immune response.
Author Response
To Reviewer 1
Manuscript ID: ijms-1954985
Type of manuscript: Review
Title: Skin barrier function and the microbiome
Thank you very much for your kind editorial letter.
We have attempted to carefully and thoroughly address all concerns raised by the editors and referees. With the help of your suggestions, we believe our manuscript has significantly improved.
The major changes are indicated by red font.
We trust that we therewith have fulfilled all the Editor’s and Reviewer’s requests.
Thank you very much for your consideration.
- Figure 1 is a great figure showing the summary of third topic, “skin microbiome and barrier function”. However, since it does not apply to all bacteria, it would be helpful to write down the corresponding bacteria as described in the microbial barrier.
I added additional details in Figure 1.
- This article demonstrates that microbiomes maintain a low pH and stimulate the expression of AMPs by secreting lipases that hydrolyze free fatty acids. It would be better if the description in figure 1 is more detailed in accordance with this content.
I added additional details in Figure 1.
- The 5th topic, “Inflammatory skin disorders and microbiome”, is good content describing the relationship between diseases and the microbiome. However, It would be better if you summarize the contents in a table and describe which species have which mechanisms and how they affect diseases.
I added Table 1.
- In the 5th topic, the readability would be better if you add 1 blank line between the sub-paragraphs by diseases.
I added 1 blank line between the sub-paragraphs.
- In line 346-356, the article describes the association between psoriasis and microbiome through Th17 immune response. However, unlike other diseases, it seems to be less mention of specific species. It would be better to mention the species if specific species are involved in Th17 immune response.
I added additional details for the specific species on Th 17 inflammation.
Thank you again for your helpful review of our article.
Sincerely,
Miri Kim, MD. PhD
Department of Dermatology, Yeouido St. Mary’s Hospital, College of Medicine, The Catholic University of Korea, #10, 63-ro, Yeongdeungpo-gu, Seoul 07345, Korea
E-mail: gimmil@naver.com

Author Response
To Reviewer 2
Manuscript ID: ijms-1954985
Type of manuscript: Review
Title: Skin barrier function and the microbiome
Thank you very much for your kind editorial letter.
We have attempted to carefully and thoroughly address all concerns raised by the editors and referees. With the help of your suggestions, we believe our manuscript has significantly improved.
The major changes are indicated by red font.
We trust that we therewith have fulfilled all the Editor’s and Reviewer’s requests.
Thank you very much for your consideration.
- Abstract, “Microbiome cosmetics can help restore skin barrier function and improve these disease” There is not enough evidence to conclude the use of cosmetics to restore the skin barrier function, it would be better not to make the conclusion here.
⇒I removed the sentence.
- Lane 30, Lane 46-48 references should be mentioned.
⇒I added references in each parts.
- There were some expressions that were not clear. For example,
(1) Lane 51 “Recent sequencing studies have extensively mapped the species inhabiting the skin 51 of various body parts, including traits such as sebaceous (glabella), moist (antecubital 52 fossa), and dry (volar forearm)”
⇒Thank you for the comment. I added additional details.
(2) Lane 75, “at the birth” should be “at birth”
⇒I changed the expression.
(3) Lane 76, it would be better to describe what the difference between born naturally and cesarean section
⇒I added additional details.
(4) Lane 113, “Recent studies have shown that the microbiome is essential for the normal structure and function of the skin barrier in mice, promoting epithelial differentiation and supporting epithelial integrity through signaling of the aryl hydrocarbon receptor of keratinocytes” it is necessary to have deeper discussion about how aryl hydrocarbon receptor signal epithelial signaling
⇒I added additional details.
(5) Lane116, “Another role of the microbiome is to secrete sphingomyelinase, which treats lamellar lipids with ceramides [13], composing up to 50% of the stratum corneum lipid structure.”
⇒I added additional details.
(6) “when breach occurs in the skin barrier and maintains symbiotic relationships with the skin. ” the meaning is not clear.
⇒I changed the expression.
(7) Lane252, “, exposure to narrow band ultraviolet (NB-UVB) light can inflect the intestinal microbiome, showing significantly increased alpha and beta diversity with vitamin D supplement”
⇒I added additional details.
(8) “These decreases in diversity can also be seen with acute exacerbations of atopic dermatitis”
⇒I added additional details.
(9) Lane 310, “Tryptophan, a metabolite of the skin microbiome, can reduce the symptoms of atopic dermatitis by binding of aryl hydrocarbon receptor to the thymic stromal lymphopoietin promoter”
⇒I deleted the sentence. Sorry for the confusion .
(10) Lane 312, “Therefore, microbiome metabolites in atopic skin can cause inflammation and maintain and worsen barrier dysfunction.” Aforementioned that tryptophan can reduce symptom of inflammation in atopic dermatitis, it is not reasonable of the sentence above.
⇒I changed the expression.
Thank you again for your helpful review of our article.
Sincerely,
Miri Kim, MD. PhD
Department of Dermatology, Yeouido St. Mary’s Hospital, College of Medicine, The Catholic University of Korea, #10, 63-ro, Yeongdeungpo-gu, Seoul 07345, Korea
E-mail: gimmil@naver.com

Round 2
Reviewer 2 Report
English editing is necessary. Lane 383-385, “However” and “although” are rarely used at the same time. Lane 267, “inflect” is not appropriate here. There are still many errors in the manuscript.
In the references section, “1” was found after the reference no. 109, please remove it.
Lane 153, using the word “two barriers” is not suitable since there are not really two barriers.
Deeper discussion about the possible pathogenesis is still lacking
Author Response
Thank you very much for your kind editorial letter.
This journal has been edited in English, but as you said, it needs further editing.
We have contacted the company for additional editing, and if necessary, we will submit revisions as soon as they arrive.
The part you mentioned has been corrected.
Thank you.
